# *Déjà vu* All Over Again: A Unitary Biological Mechanism for Intelligence Is (Probably) Untenable

**DOI:** 10.3390/jintelligence8020024

**Published:** 2020-06-02

**Authors:** Louis D. Matzel, Dylan W. Crawford, Bruno Sauce

**Affiliations:** 1Department of Psychology, Rutgers University, Piscataway, NJ 08854, USA; dylan.crawford@rutgers.edu; 2Department of Neuroscience, Karolinska Institute, 17177 Stockholm, Sweden; bruno.sauce@ki.se

**Keywords:** intelligence, processing speed, attention, working memory, heritability

## Abstract

Nearly a century ago, Spearman proposed that “specific factors can be regarded as the ‘nuts and bolts’ of cognitive performance…, while the general factor is the mental energy available to power the specific engines”. Geary (2018; 2019) takes Spearman’s analogy of “mental energy” quite literally and doubles-down on the notion by proposing that a unitary energy source, the mitochondria, explains variations in both cognitive function and health-related outcomes. This idea is reminiscent of many earlier attempts to describe a low-level biological determinant of general intelligence. While Geary does an admirable job developing an innovative theory with specific and testable predictions, this new theory suffers many of the shortcomings of previous attempts at similar goals. We argue that Geary’s theory is generally implausible, and does not map well onto known psychological and genetic properties of intelligence or its relationship to health and fitness. While Geary’s theory serves as an elegant model of “what could be”, it is less successful as a description of “what is”.

Almost a century ago, [34] ([34]) wrote that “specific factors can be regarded as the ‘nuts and bolts’ of cognitive performance, or the processes necessary for the solution to particular problems, while the general factor was the *mental energy* available to power the specific engines”. [6] ([6]) takes Spearman’s analogy of “mental energy” quite literally and doubles-down on the notion by proposing that a unitary energy source regulates both cognitive function and health-related outcomes.

[6] ([6], [7])’s hypothesis can be reduced to two statements. First, individual differences in intelligence are the consequence of the efficacy of a hierarchy of multiple cognitive and brain systems, where the efficacy of each level is dependent on the efficacy of the level below it. Geary states that “cellular energy” is the lowest common currency that drives and constrains all systems above it (namely neuronal/glial, intramodular, and intermodular functions). Thus, Geary proposes that mitochondrial function is the core mechanism that regulates higher-level systems, and thus influences all cognitive processes from which “general” intelligence emerges. The second statement in Geary’s hypothesis is that the limits placed on cellular energy by mitochondrial function account for variations not only in intelligence, but also in an array of health outcomes that co-vary with intelligence, such as age-related cognitive declines, overall physical health, and resilience against disease.

Geary’s hypothesis is well-developed, and we salute him for making specific and testable predictions and for explicitly describing potential weaknesses of his theory. This type of approach to theory construction is extremely fruitful (both in the field of intelligence and in science more generally). Here we will consider the overall plausibility of Geary’s theory and how the predictions that it generates map onto well-established facts about intelligence.

## 1. Overall Plausibility

We should first consider whether mitochondrial function is a good candidate for the “lowest common currency” that regulates higher-level systems. The answer to this question is “maybe” or “maybe not”. There is a multitude of molecular/cellular functions that are conserved across the nervous system and that could influence higher-level processes. For instance, we could easily imagine that the integrity of RNA transcription or the balance of intracellular Ca^2+^ could play an equally vital role. Both of these biological mechanisms could impact an array of cognitive processes, and both become increasingly dysfunctional as we age ([4]; [44]).

In an obscure publication 20 years ago, one of us ([19]) proposed a hypothesis that, like Geary’s, was rooted in the assumption that a “lowest common currency” could regulate the efficacy of all higher processes. At that time, we argued that several core influences on synaptic transmission might set limits on higher-order processes and could contribute to variations in general intelligence. A core mechanism particularly well-suited for this role involved the synaptic vesicle protein SV2. This protein is expressed in all (or at least most) pre-synaptic terminals in the central nervous system and determines the likelihood of transmitter vesicle binding and thus the efficacy of synaptic transmission. Such a ubiquitous protein could have a broad impact on cognitive processes, leading us to propose that SV2 levels are a critical determinant of an individual’s intelligence. For many of the same reasons discussed by Geary, this seemed a reasonable possibility. Unfortunately (for us), *evidence* got in the way of what we were convinced was an elegant idea. As we developed animal models to characterize individual differences in intelligence, we found no evidence for either differences in gene expression or protein levels of SV2 that could account for differences in general cognitive performance. Moreover, as we did behavioral experiments, it became clear that the empirical reality did not match our predictions: our smart animals did not exhibit faster reflexes (as would be a consequence of increased synaptic efficacy) and all higher-level psychological processes were not uniformly associated with variations in general cognitive ability. At first glance, it might very well be that, as Geary proposes, mitochondrial function is the lowest common currency that regulates intelligence. However, the moral of our 20-year-old tale is that many cellular functions might play a similar role, and absent critical data, it is difficult to separate what might be from what is.

Even if we granted to Geary that mitochondrial function is indeed the currency common to all biological processes relevant to cognition, it is hard to imagine that it is a frequent bottleneck. Geary assumes that any mechanism “driving the development and expression of all biological systems” necessarily “places upper-limit constraints on the development and expression” of these systems ([7]). But that assertion is not necessarily true. Energy production is unlikely to place constraints in the normal range of cognitive functioning. In eukaryotic organisms, hundreds of millions of years of co-evolution between mitochondrial DNA (mtDNA) and nuclear DNA (nDNA) shaped them to guarantee at least enough energy for typical activities targeted by natural selection ([9]; [39]). Some mammals, for example, adapted to accommodate big spikes in energy demands when hunting (such as tigers) and others adapted to utilize sustained high-energy output for long periods (such as humans, who have many adaptations to long-distance running). The human brain is energy-hungry, but even the impressive figure of 20% of the body’s overall energy expenditure (at rest) is comparable to the liver and skeletal muscles and, per gram of tissue, is much smaller than heart and kidneys ([40]). Furthermore, intense physical activities such as running, swimming, and weightlifting can consume extremely high amounts of energy per hour ([24]), with kilojoule values much above typical brain activity.

If Geary’s hypothesis is correct, it implies that mtDNA still contains enough mutations to bottleneck many typical (and critical) activities. Under that scenario, mitochondria are ill-prepared for the typical energy demands of the human brain and of every other energy-hungry organ in the body, such as skeletal muscles and the liver. Therefore, Geary’s hypothesis predicts not only a correlation of intelligence with health but also with strength, stamina, metabolic rate, digestion, hormonal balance, etc. That is possible, of course, but not so plausible.

Perhaps a bigger plausibility problem to Geary’s hypothesis is that having mitochondria as a bottleneck is in opposition to proposals that low intelligence is sometimes adaptive and related to high physical prowess (for examples and discussions on this, see [20]; [21]). For low intelligence to be adaptive, these adaptations need to be in a “less-common” currency (such as neuroarchitecture) that can isolate muscle, liver, kidney, and brain bottlenecks. Otherwise duller individuals will be worse in *everything*—hardly something that natural selection would favor!

We leave open to Geary that energy production might be a bottleneck to intelligence in very extreme (non-typical) cases such as rare mtDNA mutations that lead to mitochondrial diseases. But as we will see below, even in this later regard the literature is conflicted. Additionally, in a charitable interpretation, age declines might also be seen as extreme cases (60+ year-old individuals were outliers during our evolution, and, hence, the functioning of their mitochondria was less shaped by natural selection). So perhaps this exception from extreme cases could tie together cognitive decline with mitochondrial function, as Geary proposes. But still, mitochondrial function would not explain intelligence’s variation in the healthy and non-elderly, a condition that describes most humans that have ever lived.

Even the very quest for the lowest common currency might be misguided. There could be sources of variance that are emergent and that “exist” only at higher levels, independent of what goes on at lower levels (for more on causal emergence, see [10]). Furthermore, in complex living systems, things do not simply trickle-up unimpeded. Research in Evo-Devo, for example, shows that buffers and canalization processes reduce the relevance and impact of errors/variation in lower levels ([23]). Similarly, negative feedback processes are ubiquitous and severely limit the impact of lower-level causes, with multiple examples being well-known for decades in the fields of physiology and biochemistry.

Lastly, the foundation for Geary’s hypothesis may be on unstable grounds. It’s a common (sometimes implicit) assumption that “*g*” is a causal agent that explains the positive manifold (the positive correlation between all cognitive abilities). Historically, that has grounded many of the attempts to elucidate a unitary mechanism to describe individual differences in intelligence, including our previously mentioned SV2 hypothesis and, now, Geary’s mitochondria hypothesis. However, that assumption is itself controversial, and the *g* factor might not be based on a biological substrate (at least not in the unitary sense described by Geary).

In 1916, Thomson’s article “A hierarchy without a general factor” ([35]) was probably the first to describe alternative explanations for the emergence of *g*. Since then, numerous articles and volumes on the “*g*” factor have been published, and it is now well established that the mere presence of a general factor is not sufficient evidence for a single unitary process (such as mitochondrial efficiency). Alternatively, *g* could be a statistical construct rather than a physical entity that requires a biological substrate. Accordingly, while *g* may be a useful index to classify individuals, it need not be a top-down cause for the positive manifold.

There are a number of alternative proposals to explain *g*. The most prominent explanations are the mutualism theory (that explains the positive manifold due to initially uncorrelated cognitive domains and skills gradually interacting among themselves in an internal developmental process; [38]), the gene-environment multiplier theory (that explains the positive manifold through positive reciprocal interactions between genetically endowed intelligence and experiences (i.e., intelligence influences one’s proximal environment and that environment in turn influences one’s intelligence; [3]) and the process-overlap theory (that explains the positive manifold arising from all cognitive tasks’ reliance on multiple executive functions; [16]). More recently, Savi and collaborators proposed a network model of intraindividual growth which, in addition to explaining the positive manifold, might provide a new mathematical framework for future theories of intelligence ([29]). To the extent that any (or all) of these alternative descriptions of the positive manifold (and *g*) are valid, the century-old quest for the source of *g* is futile, and most of the oomph in Geary’s hypothesis simply vanishes.

## 2. Mapping of Specific Predictions and Implications

Having addressed the overall plausibility of Geary’s theory, we should now consider its specific predictions and implications, and how well they map onto what is known.

One implication of Geary’s hypothesis is that the closer an outcome is to the core/common influence (in this case mitochondrial energy production), the more pronounced the impact on that outcome should be, as higher-level processes are “diluted” by other influences. In this regard, variations in mitochondrial function should be more closely associated with something like reaction time than higher-level psychological processes like working memory capacity or attention. We note that, at least in principle, the effect of a lower-level influence might not *dilute*, but instead *compound* itself as it progresses upward through levels. Such a snowball trend would lead to greater impact at higher levels than at lower levels. However, this is a convoluted trend not typically seen in living systems due to the presence of canalization, negative feedback, and buffering processes (as noted above). In addition, at least to our reading, the snowballing of mitochondrial function is not an explicit property of Geary’s hypothesis.

Perhaps more than any other cognitive process, reaction time (a proxy for speed of processing) and its relationship to intelligence has been extensively studied in the past century (for a sweeping discussion, refer to [13]). Reaction time is a composite of both response time (a peripheral function) and decision time (a central process). Early in its investigation, it became very clear that composite reaction time is a poor predictor of intelligence. Not only were intelligence and composite reaction time weakly correlated, but within-subject variability in reaction time was far greater than between-subject variability. This led to the development of many procedures to separate reaction time into its two components: response time and decision time. Following such separation, it was determined that response time bore almost no relationship to intelligence, while decision time fared only slightly better, with correlations between IQ and decision time hovering in the range of *r* = 0.20 ([12]). These conclusions have been confirmed in quantitative mathematical models which demonstrate that decision-related processes account for the preponderance of the relationship between reaction time and intelligence ([30], [31]; [32]). So here at this simplest level, Geary’s hypothesis has made two erroneous predictions. First, the theory implies that both central and peripheral responses should be similarly sensitive to variations in mitochondrial function (assuming that they share similar requirements for energy, which is not necessarily the case; see below), and thus both decision and response times should be valid predictors of intelligence. Second, variations in both decision and response times should be more strongly related to intelligence than should be higher-level cognitive processes (which reside further from the core source of variance).

It may be valuable to consider the basis for the (weak) correlation between decision time and intelligence. What *if* the variations in mitochondrial function were most pronounced in the central nervous system (if for instance, central processes were more energy-hungry)? Might this explain why decision time is a better predictor of intelligence than simple response time? The answer to this most fundamental question is “probably not”. Typically, smart and dull individuals can exhibit similarly fast decision times (i.e., are equally energy capable), but dull individuals are much more *variable* in their responses and make more long-latency responses (leading to lower average times). This effect has been interpreted to reflect the impact of attention on decision time ([14]). Poor attention or attentional drift is a characteristic of dull individuals, and that leads to proportionately more long-latency responses. The problem for Geary’s hypothesis is that attention is further from the core influence (mitochondrial function) than is response time or even decision time, thus it should be more weakly related to intelligence than a more rudimentary process like reaction time. In contrast, were we to assume that decision time and attention are equally energy-hungry (or were at an equally high processing “level”), we would then expect both processes to be similarly impacted by variations in mitochondrial function. This could not be the case simply because “dull” individuals are fully capable of fast decision times comparable to those of more intelligent individuals. Regardless of the prediction (based on two interpretations of Geary’s hypothesis), the actual data appears incompatible with the theory.

What about health outcomes? Smarter individuals live longer ([42]; [43]), have less Type II diabetes ([25]), are less likely to develop cancer ([18]), and even get in fewer automobile accidents ([18]). These (as well as many other) health outcomes are reminiscent of the fact that intelligence predicts career success and satisfaction ([8]). We (the authors) have never considered these relationships to be mysterious and have never thought to explain them by alluding to common lower-level biological mechanisms. Smarter individuals simply reason more effectively, know more, and are capable of translating that knowledge into better decisions that impact health, safety, and happiness. We do not need to allude to mitochondrial function to explain why smarter individuals are less likely to use tobacco ([41]). Moreover, Geary asserts that energy production (based on the pool of mitochondria) “will be expressed throughout the body and will contribute to individual differences in resilience to disease”. So, here is a simple question: why do smarter individuals not run faster? It is well established that cellular energy production is a principal contributor to both running speed and endurance, accounting (at least in part) for the success of short-distance (and possibly marathon) runners ([26]) as well as the mammalian champions of mitochondrial energy production, the Alaskan sled dog ([2]; [22]). We know of no evidence that suggests that smarter individuals are inherently faster runners (or that sled dogs are smarter than dachshunds!). Of course, better training decisions or adherence to a training regimen might promote faster running, but this is not the basis for Geary’s predictions about the relationship between intelligence and health.

Now consider how Geary’s hypothesis maps onto what is known about the genetics of intelligence. Mothers may contribute more to environmental sources of variance (as they are often primary caregivers), but they do not contribute more nuclear genes to their offspring than do fathers, and (contrary to popular lore) intelligence is not sex-linked. Children separated from their parents at birth tend, on average, to develop intellectual capacities that are amalgams of their mothers and fathers. As such, our intelligence is correlated with that of both of our parents.

Mitochondrial DNA is inherited exclusively from the mother, which (if mitochondrial function were the basis for variations in intelligence) should leave relatively less for the father to contribute (genetically) to the cognitive capacity of his offspring. At first glance, this simple fact would be hard to reconcile with Geary’s hypothesis. However, nDNA and mtDNA interact extensively and at several levels (With the emergence of eukaryotic cells, mitochondria outsourced many of its functions to nDNA, and mtDNA lost some relevant genes ([9]). In other words, modern mitochondria contain an incomplete blueprint for the energy factory.). Depending on the allocation of these plans across nDNA and mtDNA, one could predict a greater or lesser contribution to the intelligence of offspring from the father, but certainly not a contribution that could equal that from the mother, i.e., it seems unlikely that some advantage (in the transmission of relevant genes) would not be conferred upon the mother since it is she who provides the entirety of mtDNA. Superficial support for Geary’s prediction comes from a recent study by [1] ([1]), who report that the quantity of placental mtDNA from the mother is positively correlated with childhood IQ. However, this association was observed through differences between monozygotic-monochorionic twins, forcing the conclusion that the relationship is attributable to environmental conditions in utero (likely, in part, due to the interaction of mtDNA and nDNA in the mothers leading to better placenta with more mitochondria) rather than qualitative differences in the mtDNA code inherited from the mothers. In total, depending on whether the net genetic contributions (nDNA plus mtDNA) favoring mitochondrial functioning differ between mother and father, the issue of maternal inheritance may or may not be a problem for Geary’s hypothesis. But if the contributions are the same or better from the mother (which is likely the case), it poses a problem for Geary. In any case, this surely illustrates the need for further specificity in the development of Geary’s theory.

It must also be noted that the correlation between a child’s and father’s IQ could arise from assortative mating (e.g., smart women tend to be more receptive to smart men and vice-versa). Were this the case (and to some extent it is), then a genetic contribution from the father would not be necessary to account for the correlation between the father’s and the child’s IQ, which could in principle allow for a more dominant role for maternal inheritance. However, this explanation requires two levels of correlation between father and offspring, i.e., the father’s IQ is correlated with that of the mother, and the mother is correlated (through her genetic contribution of mtDNA) with the offspring. Mathematically, this “second-level” correlation will necessarily be weaker than a first-level correlation, again resulting in the prediction that the correlation between the mother and child should be (substantially) stronger than the correlation between the father and child (which depends on the rigor of assortative mating). Again, evidence suggests (contrary to Geary’s proposal) that mothers and fathers contribute similarly to the IQs of their children.

Further, a recent study ([15]) found that while the effect on children’s IQ from non-transmitted DNA is not more important if coming from mothers, it is more important to health-related outcomes. Geary’s hypothesis links health and IQ under the same common cause of variation, so it cannot explain why “genetic nurture” from mothers matters more for health but matters the same as father’s for IQ.

Continuing on genetics, we should note that Geary’s hypothesis has one unexpected advantage: it could account for the missing heritability of intelligence better than the popular explanation focused on (additive effects of) nDNA. To identify genes that underlie variations in complex traits, modern methods like genome-wide association studies (or GWAS) have been used to examine hundreds of thousands of nDNA variants in thousands of individuals. For intelligence, nDNA variants (in combination) explain around 5% to 10% of individual differences ([17]; [28]). In contrast, family and twin studies report “genetic” influences accounting for 50–70% of individual differences in intelligence ([37]). How is it that metrics of heritability based on family resemblance can be so high and yet the sum of DNA variants detected in GWAS explain only a fraction of these differences? Where is the remaining genetic variation? This question is widely known as the problem of “missing heritability”.

To address the missing heritability problem, two of us ([27]) have suggested that high metrics of heritability can arise from gene-environment interplay, wherein, for instance, small genetic differences (and the initial phenotype that they support) are amplified as individuals make autonomous choices within their environments. Accordingly, fewer genes (or less genetic influence) than would be expected are necessary to account for high estimates of heritability. Alternatively, Geary’s hypothesis provides another solution to the missing heritability problem, as heritability emerges in part from the transmission of mtDNA (which has been ignored in GWAS of intelligence). As noted by Hudson and colleagues ([11]), early mtDNA genetic association studies were under-powered, but recent large studies have found replicable associations with specific human diseases. So, at least for diseases, there is a promise that some of the “missing heritability” for complex traits could be partially attributed to mtDNA. However, the evidence in humans is very scarce and even in model organisms, studies on the mechanisms (and implications) of mtDNA to the missing heritability has only just started ([5]). Further, we are not aware of any direct, molecular study of mtDNA’s contribution to the heritability of intelligence. Compared to other candidate solutions to the missing heritability problem, such as gene-environment interplay and epigenetic effects, Geary’s hypothesis would require a lot of theoretical and empirical work to catch up. If correct, though, Geary’s hypothesis could at least complement other solutions and we think this is an exciting prospect for the field.

Turning now towards clinical evidence, a recent study may contradict a central prediction of Geary’s hypothesis. [33] ([33]) report that children with diagnoses of mitochondrial diseases exhibit no impairments of cognitive ability if the diagnosis was not accompanied by an incidence of seizure. That is, children who were diagnosed with a mitochondrial disease and experienced no seizure events had a mean IQ score of 100, similar to age-matched controls. However, it is fair to note that we see a number of limitations with the Shurtleff et al. study. First, seizures are probably a marker of the severity of mitochondrial disease, and, as such, Geary’s hypothesis would predict that those with seizures should express lower IQ than those without. Still though, if mitochondrial function underlies variations in IQ, we would expect that even among those without seizures, those with identified mitochondrial disease should perform worse on an IQ test than those without the disease. Of course this would be true if there were a simple linear relationship between mitochondrial function and intelligence. Yet among the Shurtleff et al. sample (which admittedly, was small), those with mitochondrial disease but without seizures exhibit approximately normal IQs. If this proposed underlying relationship took another form (e.g., one that resembles a logistic or step function) it is possible that only cases of mitochondrial disease beyond some threshold (perhaps one that also results in the occurrence of seizures) would lead to impairments in cognitive function while less severe cases leave cognitive function unaffected. All of this is further complicated by at least one other study that found lower performance on a Developmental Quotient (similar to an IQ test) across all patients (regardless of seizure history) diagnosed with mitochondrial disease. Importantly, we note that results from clinical studies of mitochondria diseases inherently cannot independently support a causal relationship between mitochondrial functioning and cognitive ability, as there are a plethora of uncontrolled variables (e.g., opportunities for social interaction that may both be hindered by mitochondrial disease and influence cognitive ability) that could confound the results. Thus, all that can be definitively concluded based on the Shurtleff et al. study is that some individuals with mitochondrial disease can perform quite well on an IQ test. Although at this point the evidence is inconclusive, Geary’s proposed model seems explicit in its prediction that perturbations of the “lowest common currency” should have broad impacts on intelligence, and this appears to not necessarily be the case.

Lastly, a larger pool of mitochondria (or an increase in energy production in a fixed pool) should have a direct impact on any process that is dependent on that energy. As noted above, the (weak) relationship between reaction time and intelligence appears to be mediated by its dependence on attention and is not itself (as a proxy for “speed of processing”) a determinant of intelligence. The lower level of a mechanism like speed of processing supports the prediction that it should be more impacted by mitochondrial function than is a higher-level process like attention (assuming that deficits at lower levels do not compound through higher levels). This same argument could be applied to many sensory processes, e.g., visual or tactile discriminations. It was once believed that these sensory processes (“the avenue to higher cognition”) should be the fundamental determinant of intelligence, and in fact, the earliest intelligence test batteries (for instance, those devised by James Mckeen Cattell) were heavily weighted to measures of tactile threshold, JNDs of weights, pitch perception, and auditory and visual reaction time (for review, see [36]). As intelligence tests, these early batteries failed miserably, having no capacity to predict academic success or other relevant outcomes. It is no surprise that measures of sensory acuity were quickly abandoned as proxies for intelligence in favor of tests that assessed more high-level processes. Again, this is hard to reconcile with Geary’s hypothesis, as energy production should have a more direct (via proximity) influence on sensory function. Humbly, we note that our earlier proposal that SV2 (as well as a multitude of other “unitary biological mechanisms”) might regulate intelligence is subject to the same criticisms and caveats.

Geary’s hypothesis should serve as a role-model on how to create and polish new theories of intelligence. His theory makes precise and testable predictions and explores ideas outside the beaten track. However, as was the case for previous attempts to describe a unitary mechanism behind intelligence, tests of Geary’s theory might not prove fruitful as the theory does not match well and sometimes is even incompatible with what *is* known. *Déjà vu* all over again.

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
