# Peer review of "Déjà vu All Over Again: A Unitary Biological Mechanism for Intelligence Is (Probably) Untenable"

_jintelligence, 2020, doi:10.3390/jintelligence8020024_

Round 1

Reviewer 1 Report

jintelligence-781234

Déjà vu all over again: A unitary biological mechanism for intelligence is (probably) untenable

EVALUATION

The authors of this commentary hold that Geary’s theory lacks enough substance to deserve serious attention. They remember one fabulous failure of their own research group after proposing one hypothesis that may resemble the proposed by Geary (“it is difficult to separate what might be from what it is”).

I recommend publication without reservations. There are several references to evidence contrary to Geary’s theory that reader of this journal must know. At the end of the day, the theory could fit some already available evidence, but it is good to put on the table the contradictory evidence we already know.

Author Response

The reviewer recommended publication without revision.  We thank the review for his/her time and support.

Reviewer 2 Report

I found this comment on Geary's Thoery of mitochondiral functioning as biological mechanisms underlying g to be well balanced and both appreciative of the theoretical effort while also critical with the specific contents. I might be biased because I share the authors' secpiticism about the theoretical value regarding the theory's predictions. Nonetheless, the authors' have provided a number of well supported arguments against Geary's Theory, which renders their evaluation and conclusion regarding the theory a balanced account of Geary's theory, from my perspective.

Let me also note, at this point, that I am no expert in the field of basic biological mechanisms of mitochondria. Still, I share the authors take on the predicitons they derived from Geary's theory together with their ciritical evlaution regarding the empricial evidence contradiciting these predictions.

The only minor comment that I have concerns the incorporation of more current research on processing speed and reaction times. Studies using mathematical models (e.g. the diffusion model) support the authors claim that mainly decision related processes, more specifically the speed of information accumulation (captured by the drift rate) is related to intelligence differences (Schmitz & Wilhelm, 2016; Schubert & Frischkorn, 2020). On the second article I am a co-author and I would like to make it clear that the authors are in no way requested to cite it. This article only summarizes the state of the art in research on intelligence and reaction times and could be a source of inspiration if the authors want to include research on mathematical modeling and biological foundations of the relationship between reaction times and intelligence. Anyways, the results from this research in general further support the fact that not all aspects of reaction times are associated with intelligence, but above all decision-related processes. The authors could consider incorporating this more recent research in this area.

Apart from that I have no further comments on this commentary and found it a very nice read that will enrich the discussion of Geary's Theory and this special issue in the Journal of Intelligence.

In the spirit of openness I sign my reviews,

Gidon T. Frischkorn

References

Schmitz, F., & Wilhelm, O. (2016). Modeling Mental Speed: Decomposing Response Time Distributions in Elementary Cognitive Tasks and Correlations with Working Memory Capacity and Fluid Intelligence. Journal of Intelligence, 4(4). https://doi.org/10/gd3vrp

Schubert, A.-L., & Frischkorn, G. T. (2020). Neurocognitive Psychometrics of Intelligence: How Measurement Advancements Unveiled the Role of Mental Speed in Intelligence Differences. Current Directions in Psychological Science, 0963721419896365. https://doi.org/10/ggkz9b

Author Response

The reviewer recommended publication with minor revision.  In particular, the reviewer asked that we incorporate more recent research into our discussion of reaction time/decision time.  That new discussion has been added to our revision, and appears in Lines 170-181 of the revised manuscript.

We thank the reviewer for his/her comments and time.

Reviewer 3 Report

This paper is a critical comment on Geary’s recent hypothesis that mitochondrial function is the root cause of individual differences in intelligence and other phenotypic characteristics such as health. The authors do a great job laying bare the fundamental problems with Geary’s proposition, while being very appreciative of his effort. Their experience with their own (failed) attempt at constructing a unitary explanation for individual differences in intelligence certainly helps in this regard.

I find the comment very compelling, and I would like to see it published as soon as possible. I have only two comments that I encourage the authors to think about.

(1) The original sin

While reading the paper, I was waiting for the authors to bring up the issue of how a “g” factor can arise. After all, it is the (sometimes implicit) assumption that a common “g” factor underlying the positive manifold of intelligence test exists that has historically motivated most attempts at constructing a unitary mechanism behind individual differences in intelligence. Ultimately, most of these attempts can be understood as a search of the biological substrate of “g”, as is the case with both Geary’s and the authors’ own SV2 hypothesis.

Therefore, I was happy to see that the authors highlight the issues with “g” on p. 3. I think the assumption that “g” is causal agent for which a biological substrate must exist is an unjustified reification of a factor-analytically derived “bundle” of variance that is "g" (I am deliberately not saying “source” of variance, as this would imply causality). The authors correctly point to prominent alternative accounts of how a g factor can arise in a positive manifold in the absence of  top-down causal influences, such as mutualism. The insight that a causal-reflective explanation is only one of many equally plausible explanations for the emergence of “g” and the positive manifold is actually much older. I think Thomson’s (1916) article in the British Journal of Psychology (“A hierarchy without a general factor”) was among the first to highlight this point. Since then, numerous articles and volumes on the “g” factor have been published, and it has become common knowledge that the presence of a g factor in a positive manifold is not sufficient evidence for a single unitary process (such as mitochondrial efficiency). A bottom-up (causal-formative) and reciprocal (network) model are the two principle – more plausible – alternatives to a g-factor-type, top-down (causal-reflective) model. Given this, it is perhaps not surprising that not a single attempt at identifying a unitary biological substrate of “g” has succeeded. Thence, it is most deplorable that, more than 100 years on, there are still highly competent researchers who indulge in the same old original sin of reifying the “g” factor and wasting their time and (mitochondrial) energy in the search for a unitary substrate or mechanism.

My humble apologies for the lengthy comment – but I think these considerations deserve more attention in the manuscript. As I said, the authors do bring this issue up, but they devote only a few sentences to it on p. 3, which seems too little. It appears to me that the century-old debate surrounding the g factor and the causal processes behind it has not yet been adequately received and reflected in some circles, perhaps especially among those with a biological/neuroscientific (as opposed to psychological or psychometric) background. Please drive this point home.

(2) IQ in individuals with mitochondrial diseases as a challenge to Geary’s hypothesis

My other comment refers to the authors’ comments on p. 6 concerning the Shurtleff et al. (2018) study. I fully agree with the authors in viewing the lack of evidence that individuals with mitochondrial diseases have a (much) lower IQ as a challenge to Geary’s hypothesis. If “cellular energy” depending on mitochondrial efficiency were the root cause of variations in IQ, we would expect so see a much more skewed IQ distribution in those afflicted with mitochondrial diseases. However, it seems an overstatement to me to say that “if mitochondrial function underlies variations in IQ, we would expect that even among those without seizures, those with identified mitochondrial disease should perform worse on an IQ test than those without the disease”. For this is only true iff (sic!) IQ is a linear function of mitochondrial efficiency. It may, however, be the case that cognitive functioning is not impaired across a wide range of lower mitochondrial efficiency because (as the authors point out elsewhere) cognitive processes require only little energy – up to a tipping point beyond which cognitive functioning suffers noticeably. In this case, the relation between mitochondrial efficiency and cognitive functioning would resemble a logistic or step function. This does not invalidate the author’s point but I think they should phrase their critique more carefully here to reflect my concerns.

Of note, even if individuals with mitochondrial disease were found to have (much) lower IQs, this would not necessarily prove that low cellular energy caused by impaired mitochondrial functioning is causally responsible for cognitive impairment and low IQ. For low IQ could equally well be a result of mitochondrial effects on other developmental processes that indirectly result in lower IQ. For example with mitochondrial disease may have impaired physical mobility, which can impair the development of spatial reasoning. Equally, they may have fewer opportunities for social interactions and play in childhood, which may impair the development of verbal abilities. Such restriction on environmental stimulation during development may lead to lower performance in IQ tests even in the absence of a direct effect of mitochondrial efficiency on IQ. The authors may want to add this point to their discussion of the Shurtleff et al. (2018) study.

Overall, I applaud the authors for their efforts, and I hope my comments will help them further strengthen their commentary.

Author Response

This reviewer described our comment as "very compelling" and requested two minor revisions.

First, the reviewer asked that we more fully discuss the possibility "that a g factor can arise in a positive manifold in the absence of a top-down causal influence".  We have now added about a half of a page of further discussion of this topic, including some descriptions of the leading alternative explanations of the positive manifold.  This expanded discussion appears between lines 120-147 of the revised manuscript.

Second, regarding mitochondrial disease, the reviewer noted that we need not necessarily expect a linear function between mitochondrial efficiency and intelligence, and this could possibly explain why mitochondrial disease without seizures is not associated with impairments in IQ.  We have now conceded this as a possibility and have described the implications on Lines 297-324 of the revised manuscript.  Notably, if in fact the relationship between IQ and mitochondrial function were not linear and in fact resembled a step function, it would be hard to explain the known normalcy of the distribution of IQs, so even such nonlinearity is not consistent with Geary's hypothesis. 

We thank the reviewer for his/her time and comments.